# LEARNING RNNS WITH COMMUTATIVE STATE TRANSITIONS

## ABSTRACT

Many machine learning tasks involve analysis of set valued inputs, and thus the learned functions are expected to be permutation invariant. Recent works (e.g., Deep Sets) have sought to characterize the neural architectures which result in permutation invariance. These typically correspond to applying the same pointwise function to all set components, followed by sum aggregation. Here we take a different approach to such architectures and focus on recursive architectures such as RNNs, which are not permutation invariant in general, but can implement permutation invariant functions in a very compact manner. We first show that commutativity and associativity of the state transition function result in permutation invariance. Next, we derive a regularizer that minimizes the degree of non-commutativity in the transitions. Finally, we demonstrate that the resulting method outperforms other methods for learning permutation invariant models, due to its use of recursive computation.

## 1 INTRODUCTION

Many of the successes of deep learning can be attributed to a choice of architecture that fits the application domain. A key example of this are convolutional neural nets (Krizhevsky et al., 2012) that employ extensive parameter tying across the image, reflecting the shift invariance of visual labels. It is clear that such inductive bias is necessary for any domain if one needs to learn from a limited number of examples. This has prompted much recent interest in the question of modeling architectures that capture invariances of particular domains.

One such example is the setting where the input to the network is a set of objects and the true output does not depend on the order of the objects in the set. In this case, the function that we seek to learn is permutation invariant. It would thus be advantageous to retrict learning to models that are permutation invariant by design. This was the observation in the work on Deep Sets (Zaheer et al., 2017), which gave necessary and sufficient conditions for a network to be permutation invariant.

The underlying idea in many of the permutation invariant architectures is to apply the same network $N_1$ to all items in the set, then sum the outputs and apply another network $N_2$ to the summation. Clearly this is permutation invariant. Similarly other models using attention (Vinyals et al., 2016; Lee et al., 2019) are permutation invariant since they only rely on comparisons between elements in the set.

Although the above approaches results in invariant models, these are not clearly the smallest models that can compute a given invariant function. For example, consider the problem of computing the permutation invariant function $f(x_1, \ldots, x_n) = \max_i x_i$. In order to calculate it with a DeepSet approach, one would need to set $N_1(x) = x^k$ for large $k$ and then $N_2(z) = z^{1/k}$. This would approximate the $\ell_\infty$ metric and therefore the max function. However, in order to implement these two functions $N_1, N_2$ with ReLU networks, one would need $O(\frac{k}{\epsilon})$ to achieve $\epsilon$ accuracy (e.g., see Telgarsky, 2017, Lemma 3.4), and thus calculating this seemingly simple function to good accuracy would take a fairly large network. Clearly, if one uses max as the aggregation function in DeepSets, this could be avoided, but the same would then be true for other aggregation functions.

On the other hand, the max function can be easily implemented via a simple RNN. Consider an RNN with the following state update $s_{t+1} = \max(s_{t-1}, x_t)$ and set $s_0 = -\infty$. Then clearly we would have $s_n = \max_{i=1} x_i$. Namely, we will implement the permutation invariant max function.

Furthermore, the $s_{t+1}$ state update can be very easily be implemented using a one hidden layer ReLU network, because $max(s, x) = \text{ReLU}(s - x) + \text{ReLU}(x) - \text{ReLU}(-x)$.

The above example demonstrates that RNNs can in some cases be a natural computational model for permutation invariant functions. The goal of this paper is to ask which RNNs compute permutation invariant functions, and how they can be learned.

Clearly, most RNNs are not permutation invariant, as their state has the capability of tracking temporal patterns, and is thus sensitive to changes in ordering. Here we show that by restricting RNN state updates to commutative-associative functions, RNNs do become invariant, and the resulting set of functions is very expressive.

After establishing the commutative-associative constraints, we turn to ask how one can learn RNNs that satisfy the latter constraint. We show that for ReLU activation functions the commutative constraint corresponds to a closed form regularizer on the parameters of the RNN and suggest learning with this regularizer in order to achieve commutativity.

We show empirically that learning with commutative regularization leads to architectures that are permutation invariant in practice and that outperform DeepSets on several benchmarks.

Taken together, our results highlight the importance of the function class used for learning permutation invariant functions, and the important role of recursive computation for these tasks.

## 2 PROBLEM FORMULATION

We consider the problem of learning functions that map a sequence of inputs $\boldsymbol{x}_1, \ldots, \boldsymbol{x}_n$ to an output $y$. We note we could have also addressed the problem of generating an output sequence $y_1, \ldots, y_n$ but use a single output for simplicity.

We say that such a function $f(\boldsymbol{x}_1, \ldots, \boldsymbol{x}_n)$ is permutation invariant if for any permutation $\pi$ of $[n]$ and for all $x$ inputs we have:

$$f(\boldsymbol{x}_1, \ldots, \boldsymbol{x}_n) = f(\boldsymbol{x}_{\pi(1)}, \ldots, \boldsymbol{x}_{\pi(n)}) \tag{2.1}$$

We next ask: how should an RNN be constructed such that it is guaranteed to be permutation invariant.

We consider standard RNNs, parameterized as follows. Let $\boldsymbol{s}_t \in \mathbb{R}^r$ be the state vector at time $t$. Then the state update rule is:

$$\boldsymbol{s}_{t+1} = A\sigma\left(W\psi(\mathbf{x_t}) + \Theta \boldsymbol{s}_t\right) \tag{2.2}$$

where $\psi(\boldsymbol{x}) \in \mathbb{R}^d$ is a "pre-processing" network (of arbitrary architecture), $\sigma$ is a pointwise non-linearity such as ReLU or tanh, and $A, W, \Theta$ are matrices which transform to the appropriate dimensions, specifically, $A \in \mathbb{R}^{r \times k}$, $W \in \mathbb{R}^{k \times d}$ and $\Theta \in \mathbb{R}^{k \times r}$. We note that we add a matrix $A$ to allow the non-linearity to be taken in $\mathbb{R}^k$ such that $k \neq r$.

Finally, the output is obtained via:

$$y = g(\boldsymbol{s}_t) \tag{2.3}$$

where $g$ is any multilayer neural net (the specific architecture will not be important for our derivation).

To simplify the derivation of permutation invariance we make the assumption that the dimension of the RNN state, and the dimension of the transformed input $\psi(\boldsymbol{x})$ are the same, i.e. $r = d$.[1]

Next, we introduce the following notation for the RNN state transition:

$$\mathbf{s_{t+1}} = A\sigma\left(W\psi(\mathbf{x_t}) + \Theta \mathbf{s_t}\right) = \psi(\boldsymbol{x}_t) \circ \boldsymbol{s}_t \tag{2.4}$$

Thus, the $\circ$ operation takes two vectors in $\mathbb{R}^d$ and outputs a vector in $\mathbb{R}^d$. Namely $\circ : \mathbb{R}^d, \mathbb{R}^d \to \mathbb{R}^d$.[2]

With this notation it is clear that:

$$\boldsymbol{s}_t = (\ldots((\boldsymbol{s}_0 \circ \psi(\boldsymbol{x}_1)) \circ \psi(\boldsymbol{x}_2)) \ldots \circ \psi(\boldsymbol{x}_t)) \tag{2.5}$$

We use parentheses to emphasize that the order of applying $\circ$ does matter and these operations are neither commutative nor associative for general RNNs.

---

[1]It is always possible to achieve by padding either $\boldsymbol{s}$ or $\boldsymbol{x}$ by zeros as necessary.

[2]For the reminder of this paper we omit the $\psi$ notation as it does not affect our derivations.

## 3 PERMUTATION INVARIANT RNNS

In this section we relate properties of the state transition operator $\circ$ to the permutation invariance properties of the RNN. Without loss of generality we assume that the $\psi(\boldsymbol{x}) = \boldsymbol{x}$, since this is just a pre-processing step.

**Definition 3.1.** *An RNN is permutation invariant if the function $f(\boldsymbol{x}_1, \ldots, \boldsymbol{x}_n) = g(\boldsymbol{s}_t)$ implemented by the RNN is permutation invariant.*

We begin with basic definitions.

**Definition 3.2.** *The operator $\circ$ is commutative if for all $\boldsymbol{x}, \boldsymbol{x}'$ it holds that $\boldsymbol{x} \circ \boldsymbol{x}' = \boldsymbol{x}' \circ \boldsymbol{x}$*

**Definition 3.3.** *The operator $\circ$ is associative if for all $\boldsymbol{x}, \boldsymbol{x}', \hat{\boldsymbol{x}}$ it holds that $(\boldsymbol{x} \circ \boldsymbol{x}') \circ \hat{\boldsymbol{x}} = \boldsymbol{x} \circ (\boldsymbol{x}' \circ \hat{\boldsymbol{x}})$*

**Definition 3.4.** *If an RNN has a $\circ$ operator that is both commutative and associative we refer to it as a Commutative-Associative RNNs.*

**Theorem 3.5.** *If an RNN is Commutative-Associative then it is permutation invariant. Namely $\boldsymbol{s}_t$ does not depend on the order of $\boldsymbol{x}_1, \ldots, \boldsymbol{x}_n$.*

*Proof.* The associativity property of the $\circ$ operator implies that we can remove the parentheses in Eq. 2.5 and commutativity implies that we can then switch the order of the $\boldsymbol{x}_i$ arbitrarily. Thus we have that for all permutations $\pi$:

$$\boldsymbol{s}_t = \boldsymbol{s}_0 \circ \boldsymbol{x}_1 \circ \boldsymbol{x}_2 \ldots \circ \boldsymbol{x}_t = \boldsymbol{s}_0 \circ \boldsymbol{x}_{\pi(1)} \circ \boldsymbol{x}_{\pi(2)} \ldots \circ \boldsymbol{x}_{\pi t} \tag{3.1}$$

Therefore the state is invariant to the input order, and so is the output $y$. $\square$

A complementary question is that of universality. Namely, are the above permutation invariant RNNs sufficiently expressive to capture any permutation invariant function?

**Theorem 3.6.** *Let $f(\boldsymbol{x}_1, \ldots, \boldsymbol{x}_n)$ be a permutation invariant function. Then it can be implemented with Commutative-Associative RNNs.*

*Proof.* Since the function $f$ is permutation invariant, it follows from Theorem 2 in the DeepSets paper (Zaheer et al., 2017) that there exist two functions $\phi$ and $\rho$ such that $f(\boldsymbol{x}_1, \ldots, \boldsymbol{x}_n) = \rho\left(\sum_i \phi(\boldsymbol{x}_i)\right)$. We can now implement this architecture with a Commutative-Associative RNNs as follows: set $\psi = \phi, g = \rho$ and $\boldsymbol{x} \circ \boldsymbol{x}' = \boldsymbol{x} + \boldsymbol{x}'$. Clearly this implements the DeepSet function and is Commutative-Associative because $\circ$ is just addition. $\square$

We conclude that Commutative-Associative RNNs implement permutation invariant functions and are also sufficiently expressive to capture all permutation invariant functions.

## 4 COMMUTATIVE REGULARIZATION

In the previous section we highlighted the importance of having an RNN operator $\circ$ that is both commutative and associative. This suggests that in order to learn a permutation invariant RNN we would like to learn RNNs under the constraint that $\circ$ is associative and commutative. In this section and the remainder of the paper, we focus on the latter, namely introducing a constraint (or equivalently, regularizer) that $\circ$ is commutative.

Formally, the constraint that $\circ$ is commutative corresponds to requiring:

$$\boldsymbol{x} \circ \boldsymbol{x}' = \boldsymbol{x}' \circ \boldsymbol{x} \qquad \forall \boldsymbol{x}, \boldsymbol{x}' \in \mathbb{R}^d \tag{4.1}$$

Recall that $\circ$ depends on the matrices $A, W, \Theta$ and thus the above constraint translates to a non-linear constraint on these three matrices. However, it is not clear how to write this constraint in a way that facilitates optimization. This will be our goal in what follows.

We begin by replacing requirement 4.1 with a more convenient proxy.

**Lemma 4.1.** *If there exists a distribution, $\mathcal{D}$, over $\mathbb{R}^d, \mathbb{R}^d$ with non-zero density over its support,[3] then*

$$\mathbb{E}_{\boldsymbol{x},\boldsymbol{x}'\sim\mathcal{D}}\left[\|\boldsymbol{x}\circ\boldsymbol{x}'-\boldsymbol{x}'\circ\boldsymbol{x}\|_2^2\right] = 0 \iff \mathbb{P}\left[\boldsymbol{x}\circ\boldsymbol{x}'=\boldsymbol{x}'\circ\boldsymbol{x}, \quad \forall \boldsymbol{x},\boldsymbol{x}'\in\mathbb{R}^d\right] = 1 \tag{4.2}$$

*Proof.* The proof relies on the fact that for a continuous non-negative random variable, X, is holds that (see Section A):

$$\mathbb{E}\left[X\right] = 0 \iff \mathbb{P}[X=0] = 1 \tag{4.3}$$

We now use 4.3 in order to complete the proof. Denote by $R$ the random variable $R(\boldsymbol{x},\boldsymbol{x}') \equiv \|\boldsymbol{x}\circ\boldsymbol{x}'-\boldsymbol{x}'\circ\boldsymbol{x}\|_2^2$. Obviously, $R$ is non-negative by definition. Note also that for any pair $\boldsymbol{x},\boldsymbol{x}'$ we have $\|\boldsymbol{x}\circ\boldsymbol{x}'-\boldsymbol{x}'\circ\boldsymbol{x}\|_2^2 = 0 \iff \boldsymbol{x}\circ\boldsymbol{x}'=\boldsymbol{x}'\circ\boldsymbol{x}$. It follows that:

$$\mathbb{E}\left[R\right] = 0 \iff \mathbb{P}\left[R=0\right] = 1 \iff \mathbb{P}\left[\boldsymbol{x}\circ\boldsymbol{x}'=\boldsymbol{x}'\circ\boldsymbol{x}, \quad \forall \boldsymbol{x},\boldsymbol{x}'\in\mathbb{R}^d\right] = 1 \tag{4.4}$$

$\square$

The above lemma already gives the strong result that with probability one the network is commutative. In fact a stronger result is possible, stating that the network is commutative for *all* inputs. We only need to add the restriction of the network $A, W, \Theta$ is Lipschitz. This will hold for example, if we restrict $A, W, \Theta$ to any compact set (e.g., see similar argument in Arjovsky et al., 2017).

**Lemma 4.2.** *Assume the network defined by $A, W, \Theta$ has a Lipschitz constant bounded by $K$, and that Eq. 4.2 holds under the conditions therein. Then $\forall \boldsymbol{x}, \boldsymbol{x}'$ it holds that $\boldsymbol{x}\circ\boldsymbol{x}' = \boldsymbol{x}'\circ\boldsymbol{x}$.*

*Proof.* Denote $f(\boldsymbol{x},\boldsymbol{x}') = \|\boldsymbol{x}\circ\boldsymbol{x}'-\boldsymbol{x}'\circ\boldsymbol{x}\|_2^2$. Then $f$ also has a Lipschitz constant bounded by $L$ that is a function of $K$. Assume in contradiction that there exists $\boldsymbol{x}_0, \boldsymbol{x}_0'$ such that $f(\boldsymbol{x}_0, \boldsymbol{x}_0') = \epsilon$ for some $\epsilon > 0$. Let $\mathcal{B}(c,r)$ denote an $\ell_2$ ball centered at $c$ with radius $r$. Then because $f$ is Lipschitz it follows that for all $z \in \mathcal{B}([\boldsymbol{x}_0,\boldsymbol{x}_0'],\frac{\epsilon}{2L})$ it holds that $f(z) \geqslant \frac{\epsilon}{2}$. Let $\gamma$ denote the minimal density value at $\mathcal{B}([\boldsymbol{x}_0,\boldsymbol{x}_0'],\frac{\epsilon}{2L})$. By assumption on $\mathcal{D}$ we have that $\gamma > 0$. It therefore follows that $\mathbb{P}\left[f(X) \geqslant 0.5\epsilon\right] \geqslant \mathbb{P}\left[z \in \mathcal{B}([\boldsymbol{x}_0,\boldsymbol{x}_0'],\frac{\epsilon}{2L})\right] \geqslant 0.5\gamma\epsilon > 0$ and therefore $\mathbb{P}\left[f(X)=0\right] < 1$ and we have a contradiction with Lemma 4.1. $\square$

As a conclusion of lemma 4.1, we can express $\mathbb{E}\left[\|\boldsymbol{x}\circ\boldsymbol{x}'-\boldsymbol{x}'\circ\boldsymbol{x}\|_2^2\right]$ in terms of matrices $A, W, \Theta$ for a specific distribution and derive conditions under which it is equal to zero.

**Definition 4.3.** *Consider an RNN parameterized by matrices $A, W, \Theta$ as in 2.2 with ReLU activation.*

$$\Delta(A, W, \Theta) \equiv \mathbb{E}_{\boldsymbol{x},\boldsymbol{x}'\sim\mathcal{D}}\left[\|\boldsymbol{x}\circ\boldsymbol{x}'-\boldsymbol{x}'\circ\boldsymbol{x}\|_2^2\right] \tag{4.5}$$

Below we show that $\Delta(A, W, \Theta)$ can be expressed as a simple function of $A, W, \Theta$ when using the ReLU non-linearity.

Let $\mathcal{D}$ be a multivariate Gaussian distribution with zero mean and covariance identity. In order to calculate $\Delta(A, W, \Theta)$ we will make use of the following integral (Cho & Saul, 2009):

$$\mathbb{E}\left[\sigma(\boldsymbol{u}\cdot\boldsymbol{x})\sigma(\boldsymbol{v}\cdot\boldsymbol{x})\right] = \frac{1}{\pi}\|\boldsymbol{u}\|_2\|\boldsymbol{v}\|_2\left(\sin(\theta_{\boldsymbol{u},\boldsymbol{v}}) + (\pi - \theta_{\boldsymbol{u},\boldsymbol{v}})\cos(\theta_{\boldsymbol{u},\boldsymbol{v}})\right) \equiv g(\boldsymbol{u},\boldsymbol{v}) \tag{4.6}$$

It turns out that $\Delta(A, W, \Theta)$ is a simple function of the above $g(\boldsymbol{u},\boldsymbol{v})$ function, as stated next.

Let $\boldsymbol{w}_i$ denote the $i^{th}$ row of $W$, $\theta_i$ the $i^{th}$ row of $\Theta$ and $\boldsymbol{a}_i$ the $i^{th}$ column of $A$. Also, denote the following horizontal stacking of the vectors as

$$\boldsymbol{u}_i = [\theta_{\mathbf{i}} \quad \boldsymbol{w}_i] \quad , \quad \boldsymbol{v}_j = [\boldsymbol{w}_j \quad \theta_{\mathbf{j}}] \tag{4.7}$$

and the matrices with these as columns by $U$ and $V$ respectively.

---

[3]e.g. $\forall \boldsymbol{x}, \boldsymbol{x}' \in \mathbb{R}^d, \quad f_{\mathcal{D}}(\boldsymbol{x},\boldsymbol{x}') > 0$

**Theorem 4.4.** *Assume $\boldsymbol{x}, \boldsymbol{x}'$ are independent Gaussian vectors, each with a unit covariance. Then:*

$$\Delta(A, W, \Theta) = \sum_{i=1}^{h} \sum_{j=1}^{h} \boldsymbol{a}_i \cdot \boldsymbol{a}_j \Big( g(\boldsymbol{u}_i, \boldsymbol{u}_j) - 2g(\boldsymbol{u}_i, \boldsymbol{v}_j) + g(\boldsymbol{v}_i, \boldsymbol{v}_j) \Big) \tag{4.8}$$

*Proof.* Define the stacked vector: $\boldsymbol{z} = \begin{pmatrix} \boldsymbol{x}' \\ \boldsymbol{x} \end{pmatrix}$. The function $\Delta(A, W, \Theta)$ is given by:

$$\mathbb{E}\left[ \left\| A\sigma \left( W\boldsymbol{x} + \Theta \boldsymbol{x}' \right) - A\sigma \left( W\boldsymbol{x}' + \Theta \boldsymbol{x} \right) \right\|_2^2 \right] \tag{4.9}$$

We can write 4.9 using definitions in Eq. 4.7 as:

$$\mathbb{E}\left[ \left( A\sigma \left( U^T \boldsymbol{z} \right) - A\sigma \left( V^T \boldsymbol{z} \right) \right)^2 \right] = \mathbb{E}\left[ \sigma \left( \boldsymbol{z}^T U \right) A^T A \sigma \left( U^T \boldsymbol{z} \right) \right]$$
$$- 2\mathbb{E}\left[ \sigma \left( \boldsymbol{z}^T U \right) A^T A \sigma \left( V^T \boldsymbol{z} \right) \right] + \mathbb{E}\left[ \sigma \left( \boldsymbol{z}^T V \right) A^T A \sigma \left( V^T \boldsymbol{z} \right) \right]$$

Denoting $\tilde{\boldsymbol{u}} = \sigma(U\boldsymbol{z})$, $\tilde{\boldsymbol{v}} = \sigma(V\boldsymbol{z})$, we can now write:

$$\mathbb{E}\left[ \tilde{\boldsymbol{u}}^T A^T A \tilde{\boldsymbol{u}} - 2\tilde{\boldsymbol{u}}^T A^T A \tilde{\boldsymbol{v}} + \tilde{\boldsymbol{v}}^T A^T A \tilde{\boldsymbol{v}} \right]. \tag{4.10}$$

Letting $A^T A = Q$, we can now write the above as:

$$\mathbb{E}\left[ \sum_{i=1}^{h} \sum_{j=1}^{h} \tilde{u}_i Q_{ij} \tilde{u}_j - 2 \sum_{i=1}^{h} \sum_{j=1}^{h} \tilde{u}_i Q_{ij} \tilde{v}_j + \sum_{i=1}^{h} \sum_{j=1}^{h} \tilde{v}_i Q_{ij} \tilde{v}_j \right] \tag{4.11}$$

and more compactly

$$\sum_{i=1}^{h} \sum_{j=1}^{h} \mathbb{E}\left[ Q_{ij} \left( \tilde{u}_i \tilde{u}_j - 2\tilde{u}_i \tilde{v}_j + \tilde{v}_i \tilde{v}_j \right) \right] \tag{4.12}$$

writing back $\tilde{u}_i = \sigma(\boldsymbol{u}_i \cdot \boldsymbol{z})$ and $\tilde{v}_j = \sigma(\boldsymbol{v}_j \cdot \boldsymbol{z})$ we have

$$\sum_{i=1}^{h} \sum_{j=1}^{h} Q_{ij} \mathbb{E}\left[ \sigma(\boldsymbol{u}_i \cdot \boldsymbol{z})\sigma(\boldsymbol{u}_j \cdot \boldsymbol{z}) - 2\sigma(\boldsymbol{u}_i \cdot \boldsymbol{z})\sigma(\boldsymbol{v}_j \cdot \boldsymbol{z}) + \sigma(\boldsymbol{v}_i \cdot \boldsymbol{z})\sigma(\boldsymbol{v}_j \cdot \boldsymbol{z}) \right] \tag{4.13}$$

Using Eq. 4.6, we have that Eq. 4.13 results in the expression in Eq. 4.8. $\qquad\square$

Theorem 4.4 provides an interesting characterization of when an RNN is commutative. As mentioned earlier, an RNN is commutative iff $\Delta(A, W, \Theta) = 0$. Since the theorem gives a closed-form expression for $\Delta$ it provides a measure of non-commutativity which we will optimize in what follows. The function in Eq. 4.8 does not provide clear structual constraints on the matrices $A, W, \Theta$ but a rather elaborate dependence between their elements. The function $\Delta$ indeed achieves a value of zero for the ReLU implementation of the max function discussed in the intro, as well as (infinitely) many other commutative state transition functions.

## 5 LEARNING COMMUTATIVE RNNS

In the previous section we obtained a function $\Delta(A, W, \Theta)$ whose value reflects the degree to which the state transition operator $\circ$ is commutative.

If we could enforce the constraint $\Delta(A, W, \Theta) = 0$ we could optimize over commutative RNNs. Here we follow a standard approach to learning with constraints and add $\Delta(A, W, \Theta)$ as a regularizer multiplied by a regularization coefficient $\lambda > 0$ to whatever training loss is being optimized (e.g., squared error for regression, cross entropy for classification etc). We note that if we have training data for which zero training loss can be achieved with a commutative RNN, then the optimum of the regularized objective will be a commutative RNN regardless of the value of $\lambda$. We have indeed found this to be the case in our experiments.

# 6    RELATED WORK

In recent years, the question of invariances and network architecture has attracted considerable attention, and in particular for various forms of permutation invariances. Several works have focused on characterizing architectures that are "by–design" permutation invariant (Zaheer et al., 2017; Vinyals et al., 2016; Qi et al., 2017; Hartford et al., 2018).

While the above works address invariance for sets, there has also been work on invariance of computations on graphs (Maron et al., 2019; Herzig et al., 2018). In these, the focus is on problems that take a graph as input, and the goal is for the output to be invariant to all equivalent representations of the graph. Another approach to graph invariant computations are so called neural message passing architectures (Gilmer et al., 2017). Finally, there is also recent work on graph-based representations and their relation to graph-isomorphism (Xu et al., 2018).

Our work is a departure from these ideas in two respects. First, the typical approach is to consider an architecture that is invariant by design. Here we depart from this by considering an architecture that is not generally invariant, but can be invariant for a particular setting of its parameters. This also allows us to handle cases of "near invariance" where ordering may affect the output in some cases. Second, the typical approach is non temporal, whereas ours puts a specific emphasis on state based computation. This is in fact a natural approach for processing data streams, where many algorithms store a sketch of the data and update it as samples arrive (e.g., see Alon et al., 1999).

# 7    EXPERIMENTS

In order to demonstrate the versatile nature of commutative RNNs we evaluate our method on several tasks. For each task we compare performance to a relevant DeepSet architecture. In order to fairly compare our method to Zaheer et al. (2017) we implemented our own version of DeepSets in TensorFlow. We also explored two aggregation (i.e., pooling) methods for DeepSets: max and sum, both of which were considered in Zaheer et al. (2017).

## 7.1    SEQUENCES OF DIGITS

Perhaps the simplest suite of problems over sets are basic operations involving sets of digits, as done in Zaheer et al. (2017). We evaluate our method over three such tasks, *Sum*, *Max* and *Parity* (the latter was not used in the DeepSets paper). we show that commutative RNNs are able to capture all functions easily whereas other baselines struggle and require adaptations to the specific problem at hand. We treat the sum and max experiments as regression problems. Given a set $\{x_1, \ldots, x_n\}$ where the output, $f(\{x_1, \ldots, x_n\})$ is 1-dimensional, we apply the mean square error loss, $\left(f(\{x_1, \ldots, x_n\}) - y\right)^2$ where $y$ is the ground truth. Namely, $y = \sum_{i=1}^{n} x_i$ and $y = \max\{x_1, \ldots, x_n\}$ for the Sum and Max experiments respectively.

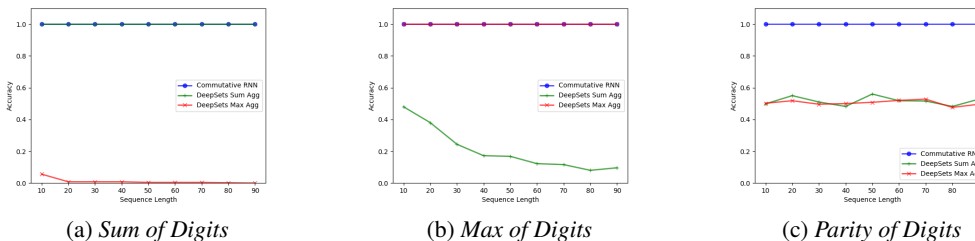

| (a) *Sum of Digits* | (b) *Max of Digits* | (c) *Parity of Digits* |

Figure 1: Experiments on digit sets. All graphs depict accuracy as a function of the sequence length. In 1a and 1b, a prediction $f(\{x_1, \ldots, x_n\})$, is considered correct if its rounded value is equal to the ground truth, i.e., $|f(\{x_1, \ldots, x_n\}) - y| \leqslant 0.5$.

### 7.1.1 Sum of Digits

We follow the exact setting described in Zaheer et al. (2017). We randomly generate $100k$ sequences of between 3 to 10 digits long where the label is the sum of the digits of the sequence. We evaluate the performance over sequences of lengths of 10 up to 100 digits long. Results are shown in Figure 1a and demonstrate that DeepSets with sum-aggregation and commutative RNNs behave similarly, as expected. However DeepSets with max aggregation does not generalize well.

### 7.1.2 Max of Digits

We perform the exact same experiment as 7.1.1 with the change that the label of a sequence is assigned with the maximal digit in the specific sequence. Results are shown in Figure. 1b and are similar to 7.1.1, except here the DeepSet sum does not generalize well.

### 7.1.3 Parity

For the parity experiment we generate $100k$ binary sequences of lengths 1 up to 10 where the assigned label is 1 if the number of 1's in the binary sequence is odd and 0 otherwise. As before, we evaluate over sequences with length up to 100. Since the parity is a binary classification problem, we apply the cross-entropy loss. Results are shown in figure 1c. Here both variants of DeepSets do not generalize well, whereas the commutative architecture has a simple solution for the parity aggregation, and therefore generalizes well.

## 7.2 Image based Sequence of Digits

MNIST8m (Loosli et al.) is a collection of 8 million $28 \times 28$ grey-scale images of digits $\{0, \ldots, 9\}$. We repeat experiments 7.1.1 and 7.1.2 where instead of sets of numerical values, we use sets of images drawn randomly to create sequences of length up to 10. We evaluate the performance of the learned models on sequences up to length 50. Figure. 2 shows that in both cases the commutative architecture performs well, whereas DeepSets only work when using the true underlying aggregation scheme.

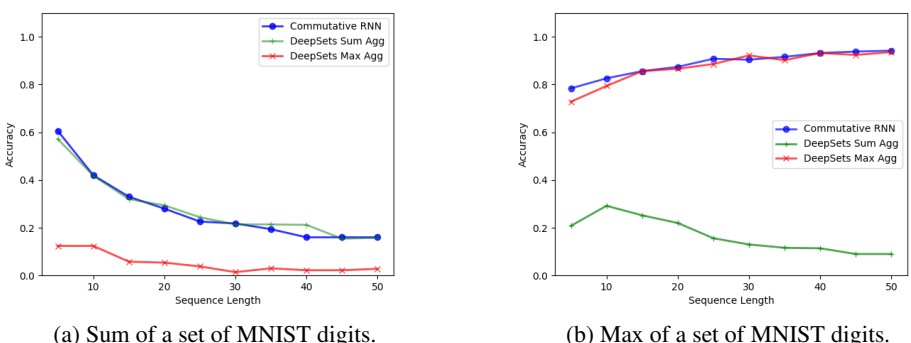

(a) Sum of a set of MNIST digits.  (b) Max of a set of MNIST digits.

Figure 2: Experiments on MNIST digits. As seen in the plots, Commutative RNNs without any adaptations perform equally to a DeepSet architecture with a modified pooling scheme.

## 8 Evaluation on ECG Recordings

We also evaluate on the ECG5000 dataset (Chen et al., 2015), which is a sequence classification problem between normal and abnormal ECG activity. There is no reason to expect this task is pertmutation invariant, but since our method only regularizes for invariance, it can also handle variant data. Results on this data for DeepSets, plain RNN and Commutative RNNs gives errors 0.9, 0.944, 0.952 respectively. Thus, indeed the regularizing effect of our methods leads to improved results on variant data as well.

Table 1: Point-cloud Classification

| Method | Accuracy |
|---|---|
| DeepSets | 0.83 |
| Commutative RNNs | 0.822 |

Table 2: Accuracy on the ModelNet40 dataset.

## 8.1 POINT CLOUD CLASSIFICATION

Point clouds are sets of 3-dimensional vectors which represent objects. Such representations of objects are useful in various fields such as robotics, computer graphics, 3D scanners, etc. The nature of point-clouds suggests that these are order-less objects and thus a natural candidate for evaluation of methods designed to handle sets. As in Zaheer et al. (2017) we use ModelNet40 Wu et al. (2015), a collection of 12k 3D representations of 40 different categories. We treat this as a 40-way classification problem. From each 3D representation we produce a point-cloud which consists of 100 3-dimensional vectors. Results are shown in Table 2, and are comparable for DeepSets and Commutative RNNs.

To summarize the empirical comparisons above, our results demonstrate that using recurrent methods to handle sets provides results that are more robust to the underlying aggregation method, and can outperform DeepSets in several settings.

## 9 DISCUSSION

We have introduced an approach to permutation invariant computation that relies on recursive architectures. While recursive architectures are generally non permutation-invariant we give conditions on their parameters such that they become invariant. The two conditions are that the state transition is associative and commutative. Of these, we focus on the latter, which we show is equivalent to minimizing a certain function of the RNN parameters. To do so we integrate the "non-commutativity" of inputs with respect to a Gaussian distribution. We believe this is a promising approach for other constraints on input space, which may be likewise represented via an appropriate measure over space.

A first natural extension of our approach is to seek a regularizer for the associative property. We explored using the Gaussian integration for this purpose but the integrals do not seem to have a closed form expression, and are related to integrating multivariate Gaussians under linear constraints, which is considered a hard problem (Miwa et al., 2003). However, it is possible that under other distributions the integral will become feasible, or that some distributions will allow faster mixing of sampling techniques.

Another extension is to other activation functions. Our derivation for the commutative regularizer relies on the structure of the ReLU function, but variants such as leaky ReLU can also be analyzed similarly. We leave analysis of other alternatives such as tanh, or more broadly LSTM like architectures for further work.

Here we consider the case of a single hidden layer state transition, although we have no constraint on the depth of the networks for $\psi(\boldsymbol{x})$ (the input processing stage) and the $g(\boldsymbol{s})$ (the state to output mapping). We note that one layer transitions are often used in practice, but it would be interesting to explore regularizers for transitions with more layers.

Another interesting learning-theoretic question is what is the sample complexity of learning with commutative and associative regularizers. For example, it might happen that in some cases the commutative constraint is sufficient to ensure a low enough sample complexity such that generalization is good even without enforcing associativity. Finally, it would be interesting to apply our approach to problems that are not "strictly permutation invariant" where regularization is expected to be more effective than a hard constraint on permutation invariance.

## A  PROOF OF EQ. 4.3.

Here we prove Eq. 4.3. Let $(\Omega, \mathcal{A}, \mathbb{P})$ be a probability space. Let $X$ be a continuous random variable such that $\forall \omega \in \Omega$ it holds that $X(\omega) \geqslant 0$ (non-negative). Assume $\mathbb{E}\big[X\big] = 0$, from Markov inequality,

$$\mathbb{P}\Big[X > \frac{1}{n}\Big] \leqslant n\mathbb{E}\big[X\big] = 0 \tag{A.1}$$

holds for any $n > 0$. To show that $\mathbb{P}\big[X = 0\big] = 1$, given $\epsilon > 0$, for any $n > \frac{1}{\epsilon}$ we have $0 = \mathbb{P}\big[X > \frac{1}{n}\big] \geqslant \mathbb{P}\big[X > \epsilon\big] \geqslant 0$, therefore $\mathbb{P}\big[X > \epsilon\big] = 0$ or equivalently, $\mathbb{P}\big[X < \epsilon\big] = 1$.

On the other hand, suppose $\mathbb{P}\big[X = 0\big] = 1$, denote by $E \subset \Omega$ the set on which $X(\omega) \neq 0$. We can write

$$\mathbb{E}\big[X\big] = \int_\Omega X(\omega)d\mathbb{P}(\omega) = \int_E X(\omega)d\mathbb{P}(\omega) + \int_{\Omega \setminus E} X(\omega)d\mathbb{P}(\omega) \tag{A.2}$$

where both terms evaluate to zero. $\int_E X(\omega)d\mathbb{P}(\omega) = 0$ since $E$ has zero measure with respect to $\mathbb{P}$ and $\int_{\Omega \setminus E} X(\omega)d\mathbb{P}(\omega) = 0$ since $\forall \omega \in \Omega \setminus E \quad X(\omega) = 0$.

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
