# OpenReview forum: "Learning RNNs with Commutative State Transitions"
_ICLR.cc/2020/Conference — Reject_

### Official Review · AnonReviewer3 · 2019-10-23
**Official Blind Review #2**

**Rating:** 1

**Review:**

Summary: this paper proposes a new principled methodology for deriving and training RNN neural networks for prediction of permutation invariant functions. Authors show on simple tasks their method may outperform DeepSets, the state of the art.


Although the idea is interesting and the paper reflects thorough work, I believe in its current form results are too weak to deserve publication. More specifically.

1)Mathematical results and statements are mostly trivial and may well be omitted or included as an appendix. They don't seem to convey anything profound (with the exception of theorem 3.6, but this follows from results on deepsets paper). Some of these results are also mostly anectodal

2)The regularization idea seems interesting, but I am concerned it is showing that the final learned networks have a deepset-like architecture: more specifically, theorem 3.6 shows RNN can implement permutation invariant functions by making identifying the parameters with the ones of deepperm. Also, as the authors mentioned, when learning a permutation invariant function then for any degree of regularization the regularization loss can be made zero. So for me, results seem to indicate that the network might have learned a deepperm kind of representation, which equivalently can be expressed as a RNN. Authors should make clear there are fundamental differents between both frameworks

3)Overall, the experimental validation section is weak and an extensive description of network architectures is lacking. Without them it is hard to resolve my concerns on 2).

**Experience Assessment:**

I have published one or two papers in this area.

**Review Assessment: Checking Correctness Of Derivations And Theory:**

I assessed the sensibility of the derivations and theory.

**Review Assessment: Checking Correctness Of Experiments:**

I assessed the sensibility of the experiments.

**Review Assessment: Thoroughness In Paper Reading:**

I read the paper at least twice and used my best judgement in assessing the paper.

---

> ### Author Response · Authors · 2019-11-13
> **Response to Reviewer #3**
>
> We thank the reviewer for the comments and points raised. We answer each point below.
>
> (1) Results are mostly trivial - we believe the results are important for context and completeness. These results should be interpreted with the limitations of DeepSets to efficiently represent certain functions such as the max function.
>
> (2) I am concerned the final learned networks have deepset-like architecture - Our network will be equivalent to a
> DeepSet only if the transition rule suffices st+1=A(W(xt)+st)=(xt)+stwhich requires very specific structure over the weight matrices. We have verified this is not the case in our experiments.
>
> (3) Extensive description on network architecture is lacking - We will add a complete description of the configurations used.

---

### Official Review · AnonReviewer2 · 2019-10-23
**Official Blind Review #2**

**Rating:** 1

**Review:**

The rebuttal did not address my concerns convincingly. There were also simple fixes that the authors could have implemented but they decided not to update the paper. I will keep my original assessment.

--------------

The premise of the work is very interesting: RNNs that are permutation-invariant. Unfortunately, the paper seems rushed and needs a better justification for not having a RNN memory that is associative. It also should cast the contributions in light of other existing work (not cited). The paper says "In this section and the remainder of the paper, we focus on the latter [commutative RNN memory operator], namely introducing a constraint (or equivalently, regularizer) that is commutative", but it never talks about the impact of a RNN memory using a non-associative operator. Being commutative is easy, isn't Equation (2.4) commutative if \Theta = W? Being associative is hard, since non-linear activations are not easily amenable to associativity.

Section 4: "The above example demonstrates that RNNs can in some cases be a natural computational model for permutation invariant functions." => Janossy pooling (Murphy et al., 2019) gives an alternative way to use RNNs, with a way to make their method tractable. Actually, my guess to why the RNNs experiments work well, even without an associative memory, is because the training examples come in multiple permuted forms, which is the data-augmentation version of the pi-SGD optimization described in Janossy pooling.

On page 1, "consider the problem of computing the permutation invariant function f(x_1, . . . , x_n) = max_i x_i", what follows is not a proof of necessity. It is an informal argument that either should be made formal or should be described as informal.

There is a lot of missing related work for sets:
Murphy, Ryan L., Balasubramaniam Srinivasan, Vinayak Rao, and Bruno Ribeiro. "Janossy pooling: Learning deep permutation-invariant functions for variable-size inputs." ICLR 2019.
Wagstaff, Edward, Fabian B. Fuchs, Martin Engelcke, Ingmar Posner, and Michael Osborne. "On the limitations of representing functions on sets." ICML 2019.
Lee, Juho, Yoonho Lee, Jungtaek Kim, Adam Kosiorek, Seungjin Choi, and Yee Whye Teh. "Set Transformer: A Framework for Attention-based Permutation-Invariant Neural Networks." ICML 2019.

Also missing related work for graphs:
Bloem-Reddy, Benjamin, and Yee Whye Teh. "Probabilistic symmetry and invariant neural networks." arXiv:1901.06082 (2019).
Murphy, Ryan L., Balasubramaniam Srinivasan, Vinayak Rao, and Bruno Ribeiro. "Relational Pooling for Graph Representations." ICML 2019.

The paper has an interesting question but needs to build on prior work. As of now, I am unconvinced that not having an associative operator for the RNN memory will lead to a good nearly permutation invariance function (unless there is data augmentation, per Janossy pooling).


**Experience Assessment:**

I have published in this field for several years.

**Review Assessment: Checking Correctness Of Derivations And Theory:**

I carefully checked the derivations and theory.

**Review Assessment: Checking Correctness Of Experiments:**

I carefully checked the experiments.

**Review Assessment: Thoroughness In Paper Reading:**

I read the paper thoroughly.

---

> ### Author Response · Authors · 2019-11-13
> **Response to Reviewer #2**
>
> We thank the reviewer for their detailed feedback and insights as well as the important remarks regarding prior work which we did not cite. Below we add clarifications to each point raised by the reviewer.
>
> (1) The authors focus on commutative without clear justification - We identify commutativity and associativity as two components of permutation invariance. In the paper we show how to regularize towards commutativity and show this is empirically effective for achieving permutation invariance. Indeed we will add quantitative evaluation to show that associativity is also achieved in these cases. Regarding setting \Theta=W, this leads to a permutation invariant network (i.e., it is associative not only commutative), but one which is less expressive and may require more parameters to fit a given function.
>
> (2) Jannosy pooling gives an alternative way to use RNNs -  Indeed Janossy gives another approach to permutation invariance using RNNs via canonical representations or sampling. We expect our approach to outperform it when sampling has high variance or the canonical representation is hard for the RNN to classify.
>
> (3) The proof of f(x_1,...,x_n)=max_i x_i does not show necessity and is informal - We will clarify this.
>
> (4) Missing work: We will add the reference and discuss relation to our approach.
>
> (5) Concerns regarding the operator not being associative - We will add an empirical evaluation of the associativity of the network as a function of its commutativity.

---

### Official Review · AnonReviewer1 · 2019-10-24
**Official Blind Review #1**

**Rating:** 3

**Review:**

The paper starts with presenting an RNN formulation and essentially writing out the sequence of RNN applications. Not surprisingly, if these applications were associative and commutative the RNN would be permutation invariant.  Then a condition for commutativity is formulated in terms of an expectation of a difference.  Based on a prior result, it is shown that the expectation can be computed in closed form.  Although it is not shown if an RNN regularized that way is permutation invariant, since the associativity is not demonstrated, empirically it is shown that it may be already of use.

   Contributions:
   1. A regularizer for RNNs that enforces commutativity
   2. A closed form for computing it
   3. A fully learnable permutation invariant "deep" network, per an empirical demonstration

   The main contribution is the empirical demonstration of the learnable nature of the obtained function unlike the prior art (e.g.  DeepSets) where a choice of the aggregation function severily affects the results.

    The theoretical component of the paper is unclear:
      1. Section 3 is rather trivial.
      2. Theorem 3.6 is hard to connect to an RNN and the rest of the paper. Unclear why bother learning the RNN at all if it needs to converge to addition of the input and hidden state to be universal anyway.
      3. In essence, the result of the paper is a way to encourage commutativity in an RNN and a demonstration that it works in practice for encouraging permutation invariance. The other explanations make things confusing and do not seem to contribute to the rest of the paper.

Could it be that the network indeed learns addition operator?  RNNs usually are only able to operate on very small sequences because of the vanishing gradient problem, yet the proposed approach will not directly work on the more robust LSTM.

Significance or lack of the difference between the proposed method and DeepSets is unclear as the plots are missing the error bars.

 The table and the accuracies reported in Section 8 are impossible to interpret. It is unclear whether the authors done cross validation.  If so, it would be helpful to see standard deviations of the reported numbers


**Experience Assessment:**

I have published one or two papers in this area.

**Review Assessment: Checking Correctness Of Derivations And Theory:**

I carefully checked the derivations and theory.

**Review Assessment: Checking Correctness Of Experiments:**

I assessed the sensibility of the experiments.

**Review Assessment: Thoroughness In Paper Reading:**

I read the paper thoroughly.

---

> ### Author Response · Authors · 2019-11-13
> **Response to Reviewer #1**
>
> We thank the reviewer for the comments and suggestions, and address specific points below.
>
> (1) Section 3 is rather trivial - we believe it is important for completeness. We will consider rephrasing.
>
> (2) Theorem 3.6 is hard to connect to an RNN and the rest of the paper - The theorem states that any permutation invariant architecture can be implemented by an associative-commutative RNN. The proof is simply to note that the DeepSet architecture is also associative-commutative. We understand this is a bit confusing, since the DeepSet is a “degenerate” RNN. Note however, that for a given permutation-invariant function (e.g. the parity in our example), there could be RNN implementations with much fewer parameters than the smallest DeepSet implementation.
>
> (3) The other explanations make things confusing - We will revise the writing style.
>
> (4) Could it be that the network indeed learns addition operator? See answer (2).
>
> (5) The approach will not work for LSTM - Indeed the regularization expression is specific to the architecture described. We agree that it would be nice to obtain a similar expression for LSTMs.
>
> (6) The plots are missing error bars - we’ll add them in future versions.
>
> (7) It’s unclear whether the authors have done cross validation - We performed cross validation, and will add the missing details and exact settings used.

---

### Decision · Program_Chairs · 2019-12-19

**Decision:**

Reject

**Comment:**

This paper examines learning problems where the network outputs are intended to be invariant to permutations of the network inputs.  Some past approaches for this problem setting have enforced permutation-invariance by construction.  This paper takes a different approach, using a recurrent neural network that passes over the data. The paper proves the network will be permutation invariant when the internal state transition function is associative and commutative.  The paper then focuses on the commutative property by describing a regularization objective that pushes the recurrent network towards becoming commutative.  Experimental results with this regularizer show potentially better performance than DeepSet, another architecture that is designed for permutation invariance.

The subsequent discussion of the paper raised several concerns with the current version of the paper. The theoretical contributions for full permutation-invariance follow quickly from the prior DeepSet results.  The paper's focus on commutative regularization in the absence of associative regularization is not compelling if the objective is really for permutation invariance.  The experimental results were limited in scope.  These results lacked error bars and an examination of the relevance of associativity. The reviewers also identified several related lines of work which could provide additional context for the results that were missing from the paper.

This paper is not ready for publication due to the multiple concerns raised by the reviewers.  The paper would become stronger by addressing these concerns, particularly the associativity of the transition function, empirical results, and related work.